# A New Microporous Lanthanide Metal–Organic Framework with a Wide Range of pH Linear Response

**DOI:** 10.3390/molecules27248696

**Published:** 2022-12-08

**Authors:** Ruyi Zhang, Liangliang Zhu, Bingbing Yue

**Affiliations:** 1School of Materials and Chemistry, Shanghai Collaborative Innovation Center of Energy Therapy for Tumors, University of Shanghai for Science and Technology, Shanghai 200093, China; 2State Key Laboratory of Molecular Engineering of Polymers, Department of Macromolecular Science, Fudan University, Shanghai 200438, China

**Keywords:** metal–organic frameworks, lanthanides, luminescence, pH response

## Abstract

Lanthanide metal–organic frameworks (Ln-MOFs) have attracted extensive attention because of their structural adjustability and wide optical function applications. However, MOFs with a wide linear pH response and stable framework structures in acidic or alkaline solutions are rare to date. Here, we used 4,4′,4″-s-triazine-2,4,6-triyltribenzoate (H_3_TATB) as an organic ligand, coordinated with lanthanide ions (Eu^3+^/Tb^3+^), and synthesized a new metal–organic framework material. The material has a porous three-dimensional square framework structure and emits bright red or green fluorescence under 365 nm UV light. The carboxyl group of the ligand is prone to protonation in an acidic environment, and negatively charged OH^−^ and ligand (TATB^3−^) have a competitive effect in an alkaline environment, which could affect the coordination ability of ligand. The luminescence degree of the framework decreases with the increase in the degree of acid and base. In particular, such fluorescence changes have a wide linear response (pH = 0–14), which can be used as a potential fluorescence sensing material for pH detection.

## 1. Introduction

Metal–organic frameworks (MOFs) are a new type of crystalline porous material that can be reasonably designed and functionalized at the molecular level, thus attracting a lot of attention [1,2,3,4,5]. Luminescence is one of the most interesting properties of MOFs. Luminescence can come from metal ions (such as lanthanide ions), organic ligands, or guest materials encapsulated inside MOFs (such as fluorescent dyes and quantum dots) [6]. Metal-centered luminescence is common in metal–organic frameworks. Among them, Lanthanide ions are photoluminescent metal ions with good luminescence performance. Due to the f-f transition and multi-level electron energy in lanthanide ions, it has the characteristics of wide emission range, clear emission band, long luminescence life, large Stokes shift, high color purity, and high photobleaching resistance [7,8].

Lanthanide metal–organic frameworks (Ln-MOFs) composed of lanthanide ions and organic ligands are an important part of luminescent MOFs. Ln-MOFs embody the unique advantages of MOFs and the inherent characteristics of lanthanide ions [9,10]. Unlike transition metal MOFs, the electrons in the f block of Ln^3+^ ions enable them to have larger coordination numbers [11]. Lanthanide ions have a high coordination number, high connectivity, and multiple coordination modes, so they have a combined advantage in terms of their ability to self-assemble into a variety of micro and nano structures, their excellent material design and processing capabilities, and their narrow and efficient luminescence [12]. Generally, due to the parity forbidden f-f transition, the molar extinction coefficient of lanthanide ions is very weak, resulting in a low quantum yield of luminescence [13,14,15]. This problem can be solved by the “antenna effect”. The ligand absorbs optical energy and then transfers it to the lanthanide ion through a molecular energy transfer mode, greatly enhancing the luminescence of lanthanide elements [16,17,18]. The energy-transfer process in Ln-MOFs is easily affected by the interaction between the host framework and the guest (such as the coordination bond, the π-π* interaction, and the hydrogen bond), resulting in a change in luminescence intensity or color [19], which has opened the door to a range of incredibly important applications [20].

Ln-MOFs have attracted extensive attention due to their unique crystal structure; fascinating photophysical properties (such as sharp emission band and high quantum yield) [21]; and potential applications in light-emitting devices [22], biomedical imaging [23,24], anti-counterfeiting labels [25,26], memory devices [27], optical fiber lasers [28], and sensors [29,30]. However, only a few MOFs have been explored as fluorescent pH sensors, especially MOF fluorescent sensors with a wide pH response of 0–14. Among them, MOFs with luminescent metal ions, pH-dependent fluorescent ligands [31], or loaded with pH sensitive fluorescent dyes [32] are excellent examples of pH sensing. According to previous reports, MOFs constructed from imidazole or pyrazole derivatives remain structurally stable under alkaline conditions [33], while carboxylate coordinated MOFs can withstand acidic environments [34,35]. So far, MOFs that can maintain a framework in both acidic and alkaline conditions have been relatively rare [36]. At present, the luminescence detection of Ln-MOFs is basically based on the mechanism of charge and energy change [37,38]. It has been documented that carboxyl oxygen atoms can be protonated under acidic conditions [39] and that negatively charged OH^−^ has a competitive effect with the ligand anion under alkaline conditions, which affects the coordination ability of the ligand and weakens the “antenna effect” of the ligand on Eu^3+^ and Tb^3+^, leading to the diminishment of luminescence [40]. Therefore, by controlling the concentration of the acid or the base, the degree of intramolecular charge transfer of the ligand can be affected, so the sensitization degree of the “antenna effect” can be effectively controlled. This provides the possibility for the pH linear sensing of luminescent MOF containing carboxyl and lanthanide metal ions with acid–base dual response.

In this paper, we prepared a new metal–organic framework by using 4,4′,4″-s-triazine-2,4,6-triyltribenzoate (H_3_TATB) as an organic ligand and coordinating with lanthanide ions (Eu^3+^/Tb^3+^). It emits bright red or green fluorescence under 365 nm UV light. It can maintain structural stability under both acidic and alkaline conditions. After soaking it in HCl or NaOH with different pH values, the luminescence decreased in varying degrees and achieved a wide range of pH linear responses (Figure 1).

## 2. Results and Discussion

Eu-TATB and Tb-TATB are prepared by 4,4′,4″-s-triazine-2,4,6-triyltribenzoate (H_3_TATB) and Ln(NO_3_)_3_·6H_2_O (Ln = Eu/Tb) in a reactor by hydrothermal method. The powder X-ray diffraction (PXRD) pattern of the prepared sample shows that its diffraction peak is narrow and strong (Figure 2a), indicating that its crystallinity is high. The obtained characteristic peaks of Eu-TATB and Tb-TATB were well matched with JCPDS cards #47-2233 and #25-1849, respectively. As shown in Figure 2b, C-O bending vibration peaks shifted from 1361 cm^−1^ (ligand) to 1359 cm^−1^ (Tb-TATB) and 1357 cm^−1^ (Eu-TATB). The bending vibration peaks of C=O shifted from 1709 cm^−1^ (ligand) to 1682 cm^−1^ (Tb-TATB) and 1658 cm^−1^ (Eu-TATB). This shows that the Eu^3+^ and Tb^3+^ coordinate with the oxygen atom on the carboxyl group of the ligand. In addition, the sample was scanned by scanning electron microscope (SEM). The sample showed a regular three-dimensional square single crystal with smooth surface (Figure 2c,d and Appendix A). Energy dispersive X-ray spectrum (EDS) mapping analysis shows that the distribution of elements in square single crystal is uniform (illustrations of Figure 2c,d).

XPS was employed to study the elemental states of the synthesized samples. The survey spectrum of Eu-TATB and Tb-TATB indicates that Eu/Tb, C, O, Na, and N are present on the particle surface (Figure 3a). Eu-TATB displayed two Eu 3d5/2 peaks at 1133.4 and 1136.3 eV that could be attributed to Eu^3+^ and O=C-O-Eu-O-, respectively. Tb-TATB displayed two Tb 3d5/2 peaks at 1240.9 and 1243.1 eV that could be attributed to Tb^3+^ and O=C-O-Tb-O-, respectively (Figure 3b), whereas the C 1s spectra was deconvoluted into three peaks: C-C/C-H (283.9 eV); C-N/C=N (285.6 eV); and C-O/C=O (287.3 eV) (Figure 3c). The O 1s is deconvoluted into two peaks: Eu-O (530.4 eV) or Tb-O (530.6 eV), and C=O (531.7 eV) (Figure 3d). The Na-O is shown with the broad photoelectron peaks at binding energies of 1071.3 eV (Na 1s) (Figure 3e). The N 1s spectra have been partitioned into two peaks: pyridinic (398.4 eV) and graphitic (401.2 eV) (Figure 3f).

Next, we studied the solid-state UV-Vis absorption spectra of H_3_TATB, Eu-TATB, and Tb-TATB (Figure 4a–c). Due to the π-π* electron transfer of the ligand, it has an absorption peak between 210 and 300 nm; 350–390 nm is the n-π* transition, and Eu-TATB has no obvious absorption peak at the corresponding position probably due to the fact that the crystallinity of Eu-TATB crystal is slightly worse than that of Tb-TATB (Appendix A), which makes it difficult to detect. When the ligand is coordinated with Eu^3+^ and Tb^3+^, the UV absorption of Eu-TATB and Tb-TATB is red-shifted, which may be due to the reduction in the steric hindrance and the enhancement of the conjugation effect after becoming a crystal. As shown in Figure 4d, Eu-TATB exhibits strong red luminescence under excitation of 350 nm, which consists of four bands at 593, 614, 652, and 700 nm, corresponding to the transition from ^5^D_0_→^7^F_J_ (J = 1, 2, 3, and 4, respectively). Tb-TATB exhibits strong green emission, and the emission peak (λ_ex_ = 350 nm) is attributed to the transition of ^7^F_J_ (J = 6, 5, 4, and 3, respectively) in the center of Tb^3+^, dominated by ^5^D_4_→^7^F_5_ emission (Figure 4g). The fluorescence microscope image shows the luminescent color and crystal morphology (Figure 4e,h).

In addition, we also studied the luminescence lifetime of Eu-TATB and Tb-TATB at room temperature (Figure 4f,i). It was found that the fluorescence lifetime of Eu-TATB at 614 nm increased from 165.4 μs to 458.6 μs compared with the raw material Eu(NO_3_)_3_·6H_2_O. Compared with the raw material Tb(NO_3_)_3_·6H_2_O, the fluorescence lifetime of Tb-TATB at 544 nm is shortened from 656.7 μs to 618.2 μs. The reason for this may be that the energy transfer efficiency of the ligand to Tb^3+^ is better than that to Eu^3+^, and the energy reverse-transfer process of the ligand to Eu^3+^ is more significant than that to Tb^3+^ [41]. This also explains why Tb-TATB can detect the n-π* transition at 350–390 nm but Eu-TATB cannot.

The crystal structure of the sample was determined by single crystal X-ray diffraction analysis. Eu-TATB and Tb-TATB crystallize in ortho-crystalline Fddd space group and have a three-dimensional framework structure (Appendix A). Its minimum asymmetric unit contains one Ln^3+^ ion, one TATB^3−^ ligand, one Na^+,^ and one formic anion (Figure 5a and Appendix A). As shown in Figure 5b and Appendix A, each Ln^3+^ ion is coordinated with eight oxygen atoms, which come from the oxygen atoms on seven carboxyl groups of seven TATB^3−^ ligands (O6, O1#1, O2#2, O5#3, O4#4, O3#5, and O4#5) and one carboxyl group of formic anion (O7), respectively. In Tb-TATB, the distance of the π-π interaction is 3.337 Å and the distance of the CH-π interaction is 2.808 Å (Figure 5c). In Eu-TATB, the distance of the π-π interaction is 3.335 Å (Appendix A) and the distance of CH-π interaction is 2.830 Å. The crystal grows further along the (100), (010), and (001) directions, eventually forming a three-dimensional framework with porous channels, characterized by carboxyl and hydrocarbon channels (Figure 5d).

### 2.1. Luminescent pH Sensing

The stability of MOFs in different chemical environments is very important for sensing applications [42]. So far, MOFs that can maintain framework structures under acidic or alkaline conditions are quite rare. Both Tb-TATB and Eu-TATB remain intact in aqueous solutions with a pH range of 0 to 14, which offer possibilities for pH sensors. Tb-TATB and Eu-TATB powder samples were soaked in HCl or NaOH aqueous solution with a pH range of 0–14 for 12 h, washed with deionized water and dried at room temperature. As shown in Figure 6a and Appendix A, the solid-state emission intensity of acid-treated samples gradually decreases with the increase in acidity and has a linear response to the pH of 0–7 (R^2^ = 0.9726 for Tb-TATB and R^2^ = 0.9652 for Eu-TATB) (Figure 6b and Appendix A). The crystal luminescence intensity immersed in pH = 7 solution is the highest. Under alkaline conditions (Figure 6c and Appendix A), the solid-state emission intensity of the sample decreases gradually with the increase in alkaline concentration, and it also has linear response to pH = 7–14 (R^2^ = 0.9443 for Tb-TATB and R^2^ = 0.9514 for Eu-TATB) (Figure 6d and Appendix A). Light microscope photos showed that the luminescence of Tb-TATB in the pH = 0–14 solution gradually weakened with the increase in the degree of acid and base, and the crystal structure remained intact and not damaged (Figure 6e). The unaltered PXRD patterns (Appendix A) of Eu-TATB and Tb-TATB also show the retention of both crystallinity and structural integrity. These materials can be used as potential pH sensors.

### 2.2. Study of Sensing Mechanism

First, we measured the lifetime of the crystals soaked in different pH solutions. The lifetime of Tb-TATB at 544 nm and that of Eu-TATB at 614 nm gradually decreased with the increase in the acid–base degree (Appendix A). This indicates that at strong acidity and alkalinity, the “antenna effect” will be affected, reducing the energy transfer between the ligand and Ln^3+^. Next, we analyzed the sites of the reaction. According to the solid-state UV-Vis absorption spectra of Tb-TATB and Eu-TATB (Appendix A), it can be found that the n-π* absorption peak corresponding to the carboxyl unit of the ligand gradually disappears at 350–400 nm with the increase in H^+^ concentration and OH^−^ concentration. It is shown that the concentration of H^+^/OH^−^ in solution affects the n-π* transition of the carboxyl group of the ligand. The carboxyl group of the ligand is prone to protonation in the acidic environment, and negatively charged OH^−^ and ligand (TATB^3−^) have a competitive effect in an alkaline environment, which affects the n-π* transition of the coordination carboxyl group and the coordination ability of the ligand, weakening the “antenna effect” and resulting in reduced luminescence. The surface areas of Tb-TATB and Eu-TATB are 158.3564 m^2^/g and 161.1914 m^2^/g, respectively. The median pore widths of the framework structures were found to be 0.7932 nm and 0.6746 nm (Appendix A), which is enough to accommodate H^+^ and OH^−^ into the framework. Thus, these two materials have appropriate pore structures, making them ideal for pH sensors.

## 3. Materials and Methods

### 3.1. Materials and General Methods

Analytically pure Eu(NO_3_)_3_·6H_2_O, Tb(NO_3_)_3_·6H_2_O, 4,4′,4″-s-triazine-2,4,6-triyltribenzoate (H_3_TATB), concentrated nitric acid, and N,N′-dimethylformamide (DMF) were commercially available and used without further purification. Powder X-ray diffraction data were collected using a Bruker D8 ADVANCE with Cu Kα radiation (λ = 1.5406 Å). Measurements were made in a 2θ range of 5–50° at room temperature with a step of 0.02° (2θ) and a counting time of 0.2 s/step. The operating power was 40 KV, 30 mA. Optical diffuse reflectance spectra were obtained on a Shimadzu UV-3600 spectrophotometer at room temperature. Data were collected in the wavelength range of 200–800 nm. BaSO_4_ powder was used as a standard (100% reflectance). Photoluminescence (PL) spectra were collected on an Edinburgh FLS-1000 luminescence spectrometer equipped with a xenon lamp. The PL decay spectra were recorded on an Edinburgh FLS-1000 luminescence spectrometer equipped with a microsecond flashlamp as the excitation source (frequency = 100 Hz) and a EPL-375 nm picosecond pulsed diode laser as the excitation source for microsecond and time-correlated single-photon counting (TCSPC) measurements, respectively. The PL lifetimes (*τ*) of solid-state samples were obtained by fitting the decay curve with a multi-exponential decay function of *I*(*t*) = *A*_1_*exp*(−*t*/*τ*1) + *A*_2_*exp*(−*t*/*τ*2) + ⋯ + *Aiexp*(−*t*/*τi*), where *Ai* and *τi* represent the amplitudes and lifetimes of the individual components for multi-exponential decay profiles. The morphology and composition of the samples were investigated by using a scanning electron microscope (SEM Hitachi S-3500, Hitachi, Ltd., Tokyo, Japan) equipped with an energy dispersive X-ray spectrum attachment (EDX Oxford Instruments Isis 300, Oxford Instruments, Abingdon-on-Thames, UK), with an acceleration voltage of 20 kV. Bright-field optical images and fluorescence microscopy images were taken from an inverted fluorescence microscope (Nikon Ti-U, Nikon, Tokyo, Japan) by exciting the samples with a mercury lamp. Nitrogen sorption experiments (up to 1 bar) for the BET surface area and porosity determination were measured with Micromeritics APSP 2460 at 77 K. About 50 to 60 mg of freshly synthesized samples were weighed before and after the degassing procedure to confirm the evacuation of the solvent.

### 3.2. Synthesis of Eu-TATB/Tb-TATB

The mixture of Eu(NO_3_)_3_·6H_2_O (0.0669 g, 0.15 mmol), H_3_TATB (0.0883 g, 0.20 mmol), and DMF and H_2_O (DMF:H_2_O = 3:2) was put into a beaker. Then, the pH of the mixture was adjusted to 4.5 using nitric acid (1 M) and sodium hydroxide (1 M) after 10 min of ultrasound. The solution was then transferred to a 20 mL teflon reactor and heated at 100 °C for 24 h. After cooling to room temperature, white crystals were collected through filtration, washed with DMF and EtOH, and successively and dried in air. The yield of Eu-TATB based on Eu(NO_3_)_3_·6H_2_O was 54.97%. The preparation method of Tb-TATB was the same as above, and the yield of Tb-TATB was 56.95%.

## 4. Conclusions

In conclusion, a novel Ln-MOFs material is assembled by 4,4′,4″-s-triazine-2,4,6-triyltribenzoate (H_3_TATB) and Eu^3+^/Tb^3+^ ions. Tb-TATB and Eu-TATB have smooth surfaces and porous three-dimensional square single crystal structures. The coordination ability of ligands could be affected in strong acids and bases, weakening the “antenna effect” between the ligand and Ln^3+^ and resulting in a decrease in luminescence intensity and a linear response with the increase in the acid–base degree. MOFs with a wide range of linear pH responses that maintain framework structure stability in acidic or alkaline solutions are rare. This study provides a new option for fluorescent visualization sensors for detecting an acid–base environment and hopes to promote the development of MOF applied materials.

## Figures and Tables

**Figure 1 molecules-27-08696-f001:**
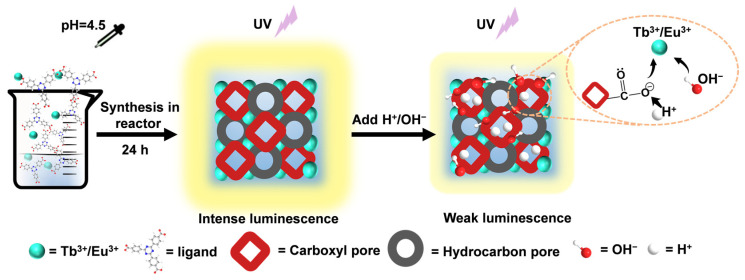
Preparation of Ln-TATB and schematic diagram of framework luminescence change behavior caused by strong acid–base environment.

**Figure 2 molecules-27-08696-f002:**
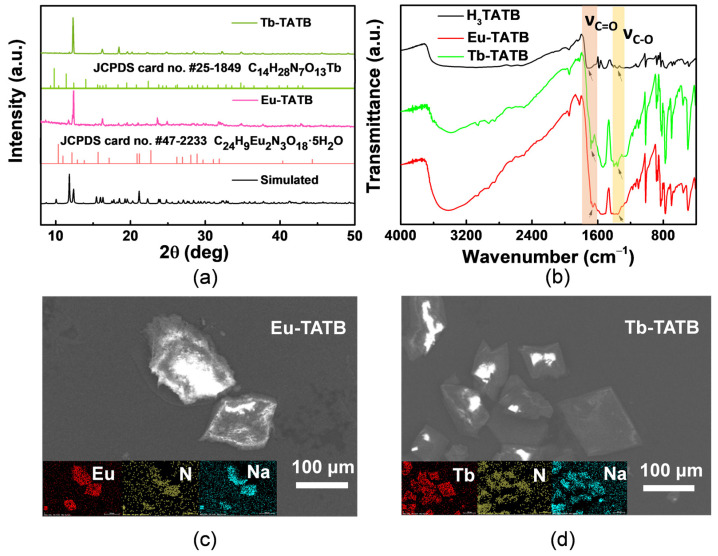
(**a**) PXRD patterns of Eu-TATB, Tb-TATB, and the simulated PRXD pattern. (**b**) FTIR spectra of Eu-TATB, Tb-TATB, and H_3_TATB. SEM images of (**c**) Eu-TATB and (**d**) Tb-TATB. Inserts show the corresponding EDS mapping.

**Figure 3 molecules-27-08696-f003:**
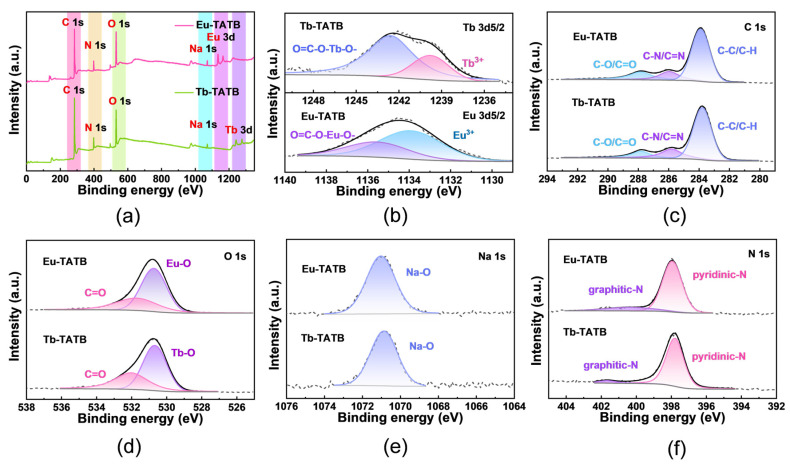
High-resolution XPS spectra of (**a**) survey scan, (**b**)Tb 3d5/2 and Eu 3d5/2, (**c**) C 1s, (**d**) O 1s, (**e**) Na 1s, and (**f**) N 1s.

**Figure 4 molecules-27-08696-f004:**
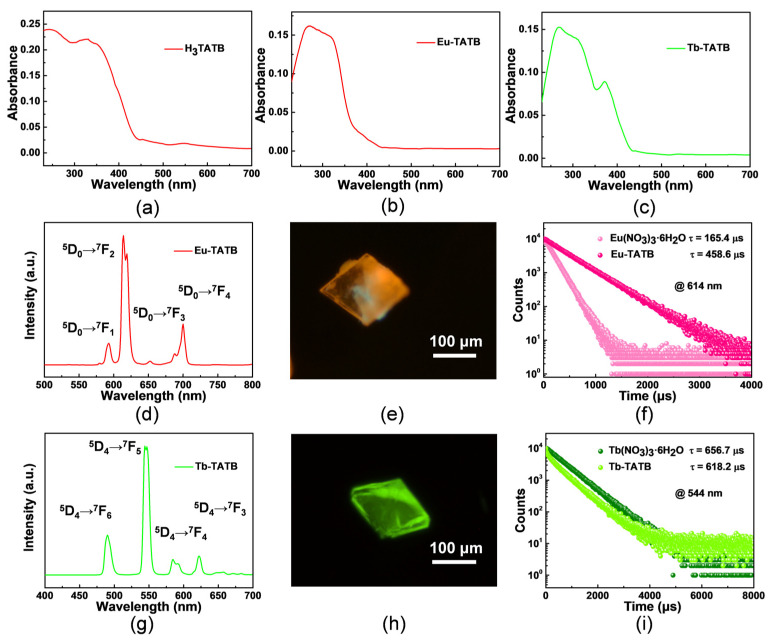
The solid-state UV-Vis absorption spectra of (**a**) H_3_TATB, (**b**) Eu-TATB, and (**c**) Tb-TATB. (**d**,**g**) Emission spectra and (**e**,**h**) photoluminescence microscope images of Eu-TATB and Tb-TATB square crystal. (**f**) PL lifetime spectra of Eu(NO_3_)_3_·6H_2_O and Eu-TATB (λ_em_ = 614 nm). (**i**) PL lifetime spectra of Tb(NO_3_)_3_·6H_2_O and Tb-TATB (λ_em_ = 544 nm).

**Figure 5 molecules-27-08696-f005:**
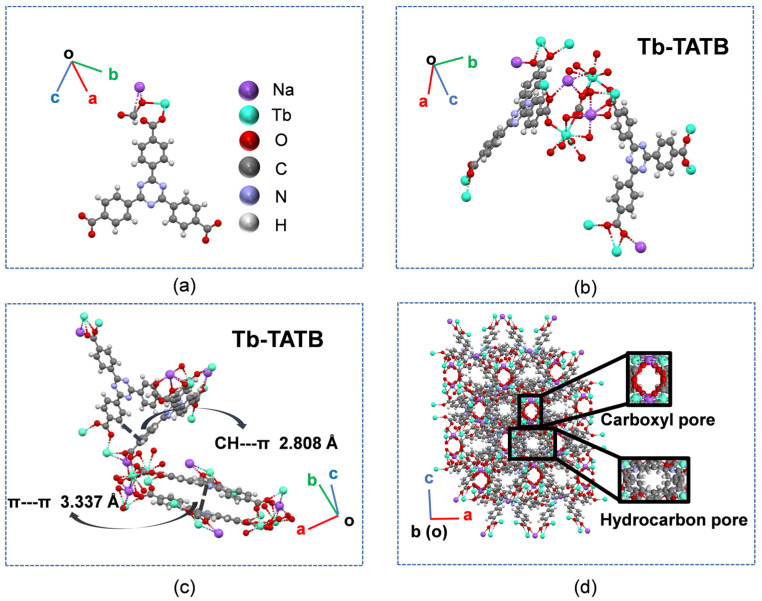
(**a**) Minimum asymmetric element structure of Tb-TATB. Deep purple = Na; lake blue = Tb; red = oxygen; gray = carbon; light purple = nitrogen; and white = hydrogen. (**b**) Schematic diagram of Tb^3+^ coordination environment. (**c**) Tb-TATB crystals. CH-π, π-π short-contacts were shown in these stacked molecular conformations. (**d**) Three-dimensional framework structure diagram of square crystal from the 001 plane.

**Figure 6 molecules-27-08696-f006:**
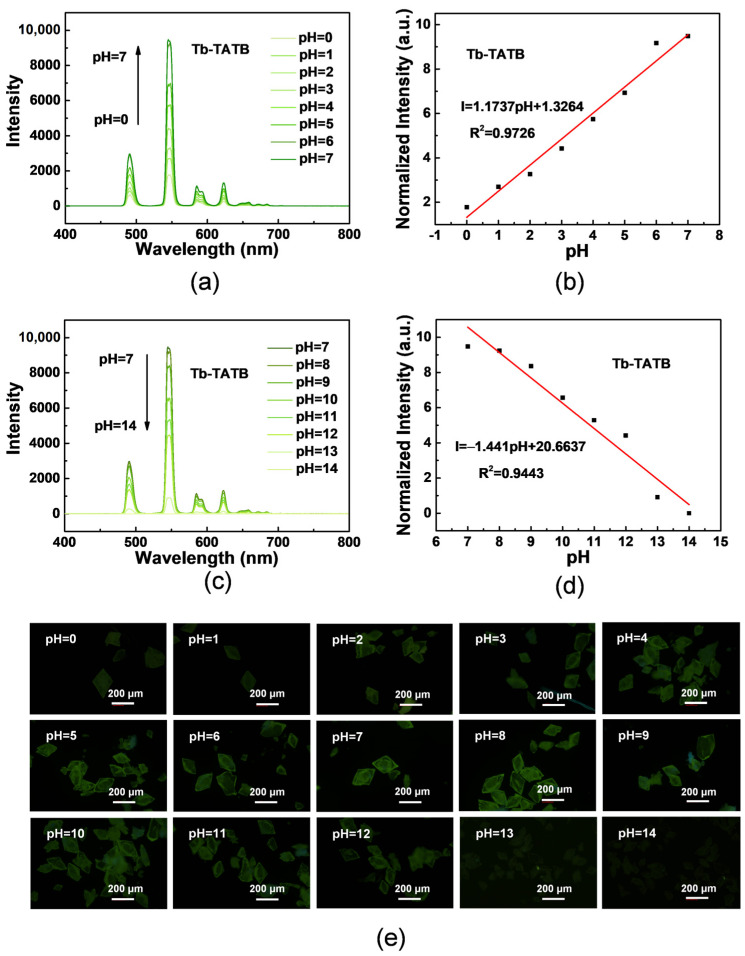
(**a**,**c**) The change in fluorescence emission intensity of Tb-TATB treated by HCl/NaOH aqueous solutions with pH ranging from 0 to 14. (**b**,**d**) The linear variation of the emission (545 nm) intensity for Tb-TATB treated by HCl/NaOH aqueous solutions with pH ranging from 0 to 14. (**e**) Fluorescence microscope images of Tb-TATB at different pH (pH = 0–14).

## Data Availability

CCDC 2217468 and 2217637 contains the supplementary crystallographic data for this paper. These data are provided free of charge by the joint Cambridge Crystallographic Data Centre www.ccdc.cam.ac.uk/structures (accessed on 15 October 2022) using the accession identifiers CCDC-2,217,468 and CCDC-2,217,637, respectively. All other data can be obtained from the authors on request.

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
