# Peer review of "A New Microporous Lanthanide Metal–Organic Framework with a Wide Range of pH Linear Response"

_molecules, 2022, doi:10.3390/molecules27248696_

Round 1
Reviewer 1 Report
This manuscript presents the synthesis, characterization and study of two lanthanide complexes. The ligand as well as some lanthanide complexes containing it, are already previously published. To my opinion there is no novelty in the present study.
The manuscript (and the supplementary material) contains two crystal structures but no cif file of the crystal structures was found, no check cif files generated from platon as well as no deposition numbers of the structures at a known database.
All these make the manuscript inappropriate for publication in Molecules journal.
Author Response
Point-by-point response to the reviewer’ comments:
Reviewer: 1
Recommendation: Reject; it appears that publication in any form would be premature at this time.
Comments:
This manuscript presents the synthesis, characterization and study of two lanthanide complexes. The ligand as well as some lanthanide complexes containing it, are already previously published. To my opinion there is no novelty in the present study.
The manuscript (and the supplementary material) contains two crystal structures but no cif file of the crystal structures was found, no check cif files generated from platon as well as no deposition numbers of the structures at a known database.
Our response: We would like to thank the reviewer for the helpful suggestions toward improvement. We wish the revised version can meet the publication requirements for Molecules.
So far, MOFs that can maintain framework in both acidic and alkaline conditions have been relatively rare, especially MOF fluorescent sensors with a wide pH response of 0-14 (J. Am. Chem. Soc. 2013, 135, 13934−13938; Biosens. Bioelectron. 2017, 87, 236-241). In addition, examples of pH-MOF sensors based on Ln3+ ions are rare (J. Mater. Chem. C. 2014, 2, 8065). The previously published papers about the ligand as well as some lanthanide complexes containing it didn't have a wide range of linear pH response properties (pH=0-14).
The crystal cif and checkcif files of two new structures are added into Supplementary File to confirm their structures. We also uploaded the crystal data to the Cambridge Crystallographic Data Centre (CCDC) and added in Table S1 and Table S4 of Supporting Information with yellow background highlight. Eu-TATB Deposition Number: 2217468. Tb-TATB Deposition Number 2217637.
Reviewer 2 Report
As the manuscript stated, Ln-MOFs have attracted extensive attention due to their fascinating photophysical properties. Ln-MOFs with wide linear response to fluorescence changes have been prepared in the manuscript, and this work is very meaningful and worth being published. However, there are still some problems to be solved before this.
1) Page 4, (h) in Fig. 2 is not marked. Please check it.
2) The sequence of “2 Result and discussion” and “3 Materials and Methods” is suggested to be interchanged.
3) Scheme 1 is recommended to be introduced in “Materials and methods”, and change the word Scheme to word Figure.
Author Response
Point-by-point response to the reviewer’ comments:
Reviewer: 2
Recommendation: Publish in Molecules after minor revisions noted.
Comments:
As the manuscript stated, Ln-MOFs have attracted extensive attention due to their fascinating photophysical properties. Ln-MOFs with wide linear response to fluorescence changes have been prepared in the manuscript, and this work is very meaningful and worth being published. However, there are still some problems to be solved before this.
Our response: We would like to thank the reviewer for the positive and significative suggestions toward improvement of our manuscript. The comments and questions have been addressed in detail below.
Q1. Page 4, (h) in Fig. 2 is not marked. Please check it.
A1: We have corrected the relevant error and also addressed it in revised manuscript (Figure 4) with yellow background highlight.
Q2. The sequence of “2 Result and discussion” and “3 Materials and Methods” is suggested to be interchanged.
A2: Thank the reviewer for full consideration and suggestions. We have interchanged the sequence of “2 Result and discussion” and “3 Materials and Methods” .
Q3. Scheme 1 is recommended to be introduced in “Materials and methods”, and change the word Scheme to word Figure.
A3: We have made corresponding modifications in revised manuscript.
Reviewer 3 Report
The manuscript “A New Microporous Lanthanide Metal-Organic Framework 2 with a Wide Range of pH Linear Response”. However, more careful discussion and analysis are needed. Therefore, I recommend this manuscript for major revision.
1. The abstract need to be improve as it does not convey any scientific message to the readers.
2. Novelty of this work should be clearly elaborated in the introduction part.
3. Introduction section need some more references on MOFs (viz. Environ. Nanotechnol. Monit. Manag. Environ., 2022, 18, 100727; 2022)
4. It is recommended to match the XRD data with JCPDS card.
5. It is recommended to incorporate SEM images of higher magnification.
6. Furthermore, BET study is mandatory to incorporate for any adsorption studies. As the porosity have great influence on it.
7. XPS results need to be added more explanations
Author Response
Point-by-point response to the reviewer’ comments:
Reviewer: 3
Recommendation: Publish in Molecules after major revision.
Comments:
The manuscript “A New Microporous Lanthanide Metal-Organic Framework 2 with a Wide Range of pH Linear Response”. However, more careful discussion and analysis are needed. Therefore, I recommend this manuscript for major revision.
Our response: We would like to thank the reviewer for the helpful suggestions toward improvement of our manuscript. The comments and questions have been addressed in detail below.
Q1. The abstract need to be improve as it does not convey any scientific message to the readers.
A1: Thanks for the reviewer's full consideration and suggestions. We have added the pH sensing mechanism of the material to the abstract to make the content more scientific (see the section of the abstract in revised manuscript with yellow background highlight).
Q2. Novelty of this work should be clearly elaborated in the introduction part.
A2: As mentioned in the 3rd paragraph of the introduction part “So far, MOFs that can maintain framework in both acidic and alkaline conditions have been relatively rare.” We also added a description about the novelty of the work in the last paragraph of the introduction “It can maintain structural stability under both acidic and alkaline conditions. After soaking it in HCl or NaOH with different pH values, the luminescence decreased in varying degrees and achieved a wide range of pH linear response.” We also addressed it with yellow background highlight.
Q3. Introduction section need some more references on MOFs (viz. Environ. Nanotechnol. Monit. Manag. Environ., 2022, 18, 100727; 2022)
A3: In the first paragraph of the introduction, we added the following two references, and used this sentence to illustrate “Metal-organic frameworks (MOFs) are a new type of crystalline porous materials that can be reasonably designed and functionalized at the molecular level, thus attracting a lot of attention [1-5].”
[4] Kaushal, S.; Pal Singh, P.; Kaur, N., Metal organic framework-derived Zr/Cu bimetallic photocatalyst for the degradation of tetracycline and organic dyes. Environ. Nanotechnol., Monit. Manage. 2022, 18, 100727.
[5] Liu, C.-Y.; Chen, X.-R.; Chen, H.-X.; Niu, Z.; Hirao, H.; Braunstein, P.; Lang, J.-P., Ultrafast Luminescent Light-Up Guest Detection Based on the Lock of the Host Molecular Vibration. J. Am. Chem. Soc. 2020, 142 (14), 6690-6697.
Q4. It is recommended to match the XRD data with JCPDS card.
A4: We sincerely thank the reviewer for the valuable suggestion. We have matched the XRD data with JCPDS card. Obtained characteristic peaks of Eu-TATB and Tb-TATB were well matched with JCPDS card no.#47-2233 and #25-1849, respectively.
Q5. It is recommended to incorporate SEM images of higher magnification.
A5: Figure S1 shows the SEM images under higher magnification. The images show that the two crystals are regular three-dimensional square single crystal with smooth surface. The length, width and height of Tb-TATB are about 129.58 nm, 80.63 nm and 9.15 nm. The length, width and height of Eu-TATB are about 45.77 nm, 69.71 nm and 8.56 nm.
Q6. Furthermore, BET study is mandatory to incorporate for any adsorption studies. As the porosity have great influence on it.
A6: Figure S7a, c are N2 adsorption and desorption curves of the two materials. We have also added relevant descriptions to the revised version of the manuscript. The specific surface area of Tb-TATB and Eu-TATB is 158.3564 m2/g and 161.1914 m2/g, respectively. The median pore width of the framework structures was found to be 0.7932 nm and 0.6746 nm, respectively (Figure S7b, d), which is enough to accommodate H+, OH- into the framework. Thus, these two materials have appropriate structure of pores, making it ideal for pH sensor (see the 1st paragraph on page 8 in revised manuscript with yellow background highlight).
Q7. XPS results need to be added more explanations.
A7: Thank the reviewer for full consideration and suggestions. We have added XPS tests of two materials and described the results in detail. The survey spectrum of Eu-TATB and Tb-TATB (Figure 3a) indicates that Eu/Tb, C, O, Na and N are present on the particle surface. Eu-TATB displayed two Eu 3d5/2 peaks at 1133.4 and 1136.3 eV that could be attributed to Eu3+ and O=C−O−Eu−O−, respectively. Tb-TATB displayed two Tb 3d5/2 peaks at 1240.9 and 1243.1 eV that could be attributed to Tb3+ and O=C−O−Tb−O−, respectively (Figure 3b). Whereas the C 1s spectra was deconvoluted into 3 peaks: C-C/C-H (283.9 eV); C-N/C=N (285.6 eV); C-O/C=O (287.3 eV) (Figure 3c). The O 1s is deconvoluted into 2 peaks: Eu-O (530.4 eV) or Tb-O (530.6 eV), C=O (531.7 eV) (Figure 3d). The Na-O is shown with the broad photoelectron peaks at binding energies of 1071.3 eV (Na 1s) (Figure 3e). The N 1s spectra has been partitioned into 2 peaks: pyridinic (398.4 eV), graphitic (401.2 eV) (Figure 3f). We also addressed it in the 1st paragraph on page 4 in revised manuscript with yellow background highlight.
Reviewer 4 Report
In this manuscript, the authors investigated Ln-MOFs materials with wide linear pH response and stable framework structures in acidic or alkaline solutions. This work is interesting and Manuscript collection also is good as well as the experimental parts clearly titles and wrote. However, some characterization and crystal data have lack. So current version is not acceptable and need deep changes. After carefully and step by step changes it can reconsider again. Suggested comments listed below:
1. As a paper contained single crystal structures, crystal cif and checkcif files of two new structures are required, so these files should be submitted for review to confirm their structures.
2. The classification on the crystal structures of the title MOFs are poor. There are some problems with formatting in Table S1 and 2. In addition, some bond lengths and bond angles are not acquired to be listed, except the bond lengths and bond angles about formed by coordination bonds, for examples, Eu-O, Tb-O and Na-O, as well as O-Eu-O, O-Tb-O and O-Na-O.
3. There are some grammatical errors, for example, “Ln-MOFs sensitize lanthanide ions by strong absorption of organic ligands, namely "antenna effect", so as to greatly enhance the luminescence of lanthanide elements.” “...a new metal-organic framework was synthesized by using 4,4′,4″-striazine-2,4,6-triyltribenzoate (H3TATB) as organic ligand and coordinating with lanthanide ions (Eu3+/Tb3+).”
4. At different pH, PXRD patterns are acquired to confirm the structure stability of Tb-TATB and Eu-TATB. In particular, when the pH of the aqueous solution is 1 or 2 and 13 or 14, the PXRD patterns should be provided.
5. As discussing about the sensing mechanism, the curves in Figure S5 are not enough to illustrate the phenomena, “the n-π* absorption peak corresponding to the carboxyl unit of the ligand gradually disappears at 350 nm-400 nm with the increase of H+ concentration and OH- concentration” in the pH range of 0-7 and 7-14, respectively. Because there are only two or three curves in pH=0-7 and 7-14, respectively.
In addition, this study belongs to the field of inorganic coordination chemistry instead of organic chemistry.
Author Response
Point-by-point response to the reviewer’ comments:
Reviewer: 4
Recommendation: Publish in Molecules after carefully and step by step changes.
Comments:
In this manuscript, the authors investigated Ln-MOFs materials with wide linear pH response and stable framework structures in acidic or alkaline solutions. This work is interesting and Manuscript collection also is good as well as the experimental parts clearly titles and wrote. However, some characterization and crystal data have lack. So current version is not acceptable and need deep changes. After carefully and step by step changes it can reconsider again. Suggested comments listed below:
Our response: We would like to thank the reviewer for the positive and significative suggestions toward improvement of our manuscript. The comments and questions have been addressed in detail below.
Q1. 1. As a paper contained single crystal structures, crystal cif and checkcif files of two new structures are required, so these files should be submitted for review to confirm their structures.
A1: Thanks for the reviewer's helpful suggestions. We have added the crystal cif and checkcif files of two new structures in the Supplementary File to confirm their structures.
Q2. The classification on the crystal structures of the title MOFs are poor. There are some problems with formatting in Table S1 and 2. In addition, some bond lengths and bond angles are not acquired to be listed, except the bond lengths and bond angles about formed by coordination bonds, for examples, Eu-O, Tb-O and Na-O, as well as O-Eu-O, O-Tb-O and O-Na-O.
A2: We have corrected the formatting problems in Table S1 to Table S6 and removed some bond lengths and bond angles which are not acquired to be listed.
Q3. There are some grammatical errors, for example, “Ln-MOFs sensitize lanthanide ions by strong absorption of organic ligands, namely "antenna effect", so as to greatly enhance the luminescence of lanthanide elements.” “...a new metal-organic framework was synthesized by using 4,4′,4″-striazine-2,4,6-triyltribenzoate (H3TATB) as organic ligand and coordinating with lanthanide ions (Eu3+/Tb3+).”
A3: We have corrected some grammatical errors in the text and addressed these sentences with yellow background highlight (see the sentence in the 2nd paragraph and the first sentence in last paragraph of the introduction in manuscript).
Q4. At different pH, PXRD patterns are acquired to confirm the structure stability of Tb-TATB and Eu-TATB. In particular, when the pH of the aqueous solution is 1 or 2 and 13 or 14, the PXRD patterns should be provided.
A4: Thank the reviewer for full consideration and suggestions. We have added PXRD patterns of Tb-TATB/Eu-TATB after soaked in strong acid (pH=0-2) and base (pH=12-14) solutions for 12 hours (Figure S4). The unaltered PXRD patterns of Eu-TATB and Tb-TATB also show the retention of both crystallinity and structural integrity and prove that this material can be used as a potential pH sensor.
Q5. As discussing about the sensing mechanism, the curves in Figure S5 are not enough to illustrate the phenomena, “the n-π* absorption peak corresponding to the carboxyl unit of the ligand gradually disappears at 350 nm-400 nm with the increase of H+ concentration and OH- concentration” in the pH range of 0-7 and 7-14, respectively. Because there are only two or three curves in pH=0-7 and 7-14, respectively.
A5: We have added several data sets to better illustrate the sensing mechanism: “the n-π* absorption peak corresponding to the carboxyl unit of the ligand gradually disappears at 350 nm-400 nm with the increase of H+ concentration and OH- concentration.”
Round 2
Reviewer 1 Report
As I can see the authors describe the system as containing the main Tb complex, a part of disordered formic anion, a part of disordered sodium cation and, I suppose, a disordered part of a water molecule (most probably coordinated to Tb cation). In order this nice structure to have also a physical meaning, the formic anion and the sodium cation should have the same occupation factors. So I suggest that authors refine again with the proper occupation factors, revise the structure deposited and all tables in the manuscript. Then the manuscript will be ready for publication.
Author Response
Point-by-point response to the reviewer’s comments:
Reviewer: 1
Recommendation: Publish in Molecules after minor revisions noted.
Comments:
As I can see the authors describe the system as containing the main Tb complex, a part of disordered formic anion, a part of disordered sodium cation and, I suppose, a disordered part of a water molecule (most probably coordinated to Tb cation). In order this nice structure to have also a physical meaning, the formic anion and the sodium cation should have the same occupation factors. So I suggest that authors refine again with the proper occupation factors, revise the structure deposited and all tables in the manuscript. Then the manuscript will be ready for publication.
Our response: Thanks for the reviewer's full consideration and suggestions. The crystal structure is mentioned in the first paragraph of page 6 with yellow background highlight “Its minimum asymmetric unit contains one Ln3+ ion, one TATB3- ligand, one Na+ and one formic anion. Each Ln3+ ion is coordinated with eight oxygen atoms, which come from the oxygen atoms on seven carboxyl groups of seven TATB3- ligands (O6, O1#1, O2#2, O5#3, O4#4, O3#5, O4#5) and one carboxyl group of formic anion (O7), respectively.” In addition, it can be seen from the crystal structure diagram of Figure 5a, b and S2a, b that formic anion and sodium ions participate in the coordination. The bond lengths and bond angles about coordination bonds can be viewed in Table S2, S3, S5 and S6 in the revised supporting information.

Reviewer 4 Report
1. Except for bond lengths and bond angles about coordination bonds must be listed in supporting information, others do not be required listed in supplementary. It is suggested that people who understand crystal structures should organize this part.
2. The cif files and checkcif files of two MOFs in the attachment are complete and satisfactory.
3. The PXRD patterns of two MOFs in Figure S4 are right and in accordance with the author's conclusion.
4. In Figure S3, the conclusion, "The linear variation of the emission (614 nm) intensity for Eu-TATB..." is inappropriate for the poor R2 value and data.
Author Response
Point-by-point response to the reviewer’s comments:
Reviewer: 4
Recommendation: Publish in Molecules after minor revisions noted.
Comments:
Q1. Except for bond lengths and bond angles about coordination bonds must be listed in supporting information, others do not be required listed in supplementary. It is suggested that people who understand crystal structures should organize this part.
A1: Thanks for the reviewer's helpful suggestions. We have checked and modified the tables in the supplementary again and removed some bond lengths and bond angles which are not acquired to be listed.
Q2. In Figure S3, the conclusion, "The linear variation of the emission (614 nm) intensity for Eu-TATB..." is inappropriate for the poor R2 value and data.
A2: We have tested the emission spectra of Eu-TATB at different pH (pH=0-7) again and obtained satisfactory data.
